# Biological Implications of MicroRNAs as Regulators and Biomarkers of Therapeutic Toxicities in Breast Cancer

**DOI:** 10.3390/ijms241612694

**Published:** 2023-08-11

**Authors:** Raza Abbas Syed, Matthew G. Davey, Vinitha Richard, Nicola Miller, Michael J. Kerin

**Affiliations:** Discipline of Surgery, Lambe Institute for Translational Research, University of Galway, H91 YR71 Galway, Ireland; mattgdavey1@gmail.com (M.G.D.);

**Keywords:** microRNA, breast cancer, diabetes, cardiovascular disease, miRNA profiling, post-transcriptomic profiling

## Abstract

Contemporary breast cancer management includes surgical resection combined with a multimodal approach, including chemotherapy, radiotherapy, endocrine therapy, and targeted therapies. Breast cancer treatment is now personalised in accordance with disease and host factors, which has translated to enhanced outcomes for the vast majority of patients. Unfortunately, the treatment of the disease involves patients developing treatment-induced toxicities, with cardiovascular and metabolic side effects having negative implications for long-term quality-of-life metrics. MicroRNAs (miRNAs) are a class of small non-coding ribonucleic acids that are 17 to 25 nucleotides in length, which have utility in modifying genetic expression by working at a post-transcriptional cellular level. miRNAs have involvement in modulating breast cancer development, which is well described, with these biomarkers acting as important regulators of disease, as well as potential diagnostic and therapeutic biomarkers. This review focuses on highlighting the role of miRNAs as regulators and biomarkers of disease, particularly in breast cancer management, with a specific mention of the potential value of miRNAs in predicting treatment-related cardiovascular toxicity.

## 1. Introduction

One in eight women worldwide will be diagnosed with breast cancer [1], with the disease representing the most common cause of cancer-related death in women [2]. Therefore, it is imperative that further translational research efforts focus on adding novel diagnostic and therapeutic tools to the clinician’s armamentarium. Breast cancer is typically classified into four intrinsic subtypes, including Luminal A, Luminal B, Erb-B2 overexpressing, and “basal-like” [3]. The type of cancer diagnosed has implications in treatment selection, as well as predicting disease recurrence. Improvements in our understanding of the molecular intricacies of the disease [4], coupled with the advent of targeted therapeutics [5], have facilitated enhanced oncological and survival outcomes for the vast majority (5-year survival increasing from 62.5% in the 1970s to 86.2% in 2007, as evident from data from the Surveillance, Epidemiology and End Results (SEER) database [6]). Contemporary breast cancer treatment now includes a multimodal approach, where surgical resection is coupled with radiotherapy, chemotherapy, and endocrine agents, as well as targeted therapies (including immunomodulatory drugs) [7,8]. Moreover, surgical strategy has become more conservative in recent decades, as evidenced by Fisher et al., who previously demonstrated no significant survival difference after 20 years of follow-up in women with stage I or II breast cancer who underwent a mastectomy and those who underwent breast-conserving surgery (BCS) with or without radiation therapy post-operatively [9], which, in turn, is associated with enhanced psychosocial well-being and improved survivorship for patients [10].

Unfortunately, treatment strategies used routinely in breast cancer management are associated with several treatment-induced toxicities. For example, the cardiotoxicity from anthracycline-based chemotherapy was first reported in the late 1970s, with case reports revealing pericarditis, heart failure, and myocardial infarction occurring after administration of these chemotherapeutic compounds [11]. Furthermore, radiotherapy has also been found to negatively impact cardiac function, with the risk of ischaemic heart disease increasing in women who received radiotherapy for breast cancer treatment [12]. Trastuzumab, an important targeted therapy for HER-2/*neu*-positive breast cancer [5], is associated with significant cardiac disease, including heart failure and cardiomyopathies [13,14]. With the improved relative survival observed in patients previously treated for breast cancer in recent decades [6], it is imperative that an oncologist considers the long-term quality of life and the toxicities brought about by such treatment strategies.

MicroRNAs (or miRNAs) are a recently discovered class of ribonucleic acids (RNAs) that have been shown to be dysregulated in human breast cancer [15,16]. These small, non-coding RNAs are important regulators of gene expression, which act at a post-transcriptional level to influence protein synthesis [17,18]. In 2005, the seminal work of Iorio et al. demonstrated the presence of 29 unique miRNAs with aberrant expression in breast tumours versus normal breast tissue, revealing an exciting potential for their use in breast cancer management [15]. These miRNAs were upregulated, highlighting their role as potential oncogenes, or downregulated, displaying possible tumour suppressor activity [15]. Since the seminal work of these authors, the role of miRNAs in breast cancer has been further elucidated. For example, previous authors have demonstrated the capabilities of miRNAs to predict breast cancer hormone receptor status, others have evaluated the role of miRNAs as therapeutic options in cancer [19,20,21,22], and several previous reports have correlated miRNA expression profiles with long-term survival outcomes in malignancy [23,24,25]. 

Accordingly, the aim of this review is to highlight the modulatory role of miRNAs in breast cancer while outlining the role of miRNAs in identifying treatment-induced cardiotoxicities to conventional breast cancer management strategies and examining for a cross-link between miRNAs involved in cardiovascular disease and diabetes mellitus diagnosed secondary to breast cancer treatment. 

## 2. Biogenesis of miRNAs

miRNAs are a relatively recent discovery, with the first example, *lin-4*, described in 1993, isolated from the *C*. *elegans* nematode. [26]. These molecules can be defined as non-coding, endogenous, conserved RNA sequences that are typically between 17 and 25 nucleotides in length, which act as post-transcriptional key regulators of gene expression [17,18]. miRNAs have been shown to possess numerous roles, for example, in controlling cell proliferation and apoptosis, metabolism, and tissue differentiation [27,28,29]. With up to 1,900 human miRNA sequences currently known [30,31,32,33,34,35], they represent a growing area of study (miRBase v.22.1 accessed on 25 July 2023).

miRNAs are produced from stem–loop precursors, where RNA polymerase II transcribes these precursors to form primary precursor miRNA [36]. The stem–loop is cleaved asymmetrically within the nucleus by RNase III Drosha and the cofactor associated with this molecule, DGCR8. This process produces precursor miRNA [36], which typically measures 70 nucleotides long. The cytoplasmic transport of precursor miRNA is then mediated by exportin-5 alongside the nuclear protein Ran-GTP [37]. This exported precursor miRNA undergoes cleavage by Dicer to form duplex molecules [38,39,40,41]. The partner protein to Dicer, TRBP (TAR RNA-binding protein), associates with the Dicer endonuclease but may not be necessary in humans [42]. Dicer may also associate with the cofactor PACT (interferon-induced protein kinase) [43]. Following this, RNA duplexes are loaded onto Argonaute proteins, forming the RNA-induced silencing complex (RISC) [44]. MRNA (messenger RNA) is subsequently suppressed by RISC via base pairing to mRNA and resultant mRNA degradation based on complementarity [45]. 

## 3. miRNAs as Biomarkers of Breast Cancer

There are several various classes of biomarkers with clinical applications. For example, diagnostic biomarkers are employed to detect the presence or absence of disease in a patient [46]. Monitoring biomarkers may be used to detect the “status of a disease or medical condition, or for evidence of exposure to a medical product or environmental agent” [47]. Response biomarkers show a biological result has occurred in an individual due to a particular exposure [48], whereas predictive biomarkers try to ascertain outcomes due to particular exposures [49]. Existing biomarkers include basic blood tests, such as HbA1c levels [50], and also imaging techniques, with scoring systems such as BI-RADS (Breast Imaging Reporting and Data System) serving as examples [51]. 

The ideal biomarker should possess a number of properties, including early detection, high sensitivity, accessibility, appropriate stability, and translatability, and have an understood mechanism [52]. Mitchell et al. demonstrated that epithelial-tumour-derived miRNAs can reach plasma in levels sufficient to be detectable, with an increase observed in tumour-specific miRNAs rather than a generalised increase in total miRNA expression, indicating the suitability of miRNAs for cancer detection. This study highlighted the ability to identify prostate cancer patients from healthy controls based on miRNA-141 levels and demonstrated the inherent stability of miRNAs in plasma [53]. The long-term stability of miRNAs is demonstrated by their ability to remain detectable in formalin-fixed paraffin-embedded samples compared to RNA, making them useful for retrospective studies [54]. Heneghan et al. identified increased miRNA detection in whole blood samples compared to serum or plasma samples of breast cancer patients [55]. This, alongside the availability of powerful quantitative tools and the absence of traditional proteomic bottlenecks (such as the lack of polymerase chain reactions), marks miRNAs as potentially excellent biomarkers of disease [53,56].

In breast cancer, miRNAs were effective in predicting hormone receptor status, whereby a previous study by Lowery et al. illustrated six miRNAs that were predictive of oestrogen receptor status (miRNA-342, miRNA-299, miRNA-217, miRNA-190, miRNA-135b, and miRNA-218), four miRNAs that were predictive of progesterone receptor status (miRNA-520g, miRNA-377, miRNA-527-518a, and miRNA-520f-520c), and five miRNAs that were predictive of human epidermal growth factor receptor status (miRNA-520d, miRNA-181c, miRNA-302c, miRNA-376b, and miRNA-30e) [57]. This illustrates the potential of miRNAs to accurately molecularly subtype within breast cancer [58]. Moreover, miRNAs may potentially be used as diagnostic biomarkers, with Khalighfard et al. showing the upregulation of miRNA-21, miRNA-10b, and miRNA-155 in breast cancer patients compared with controls. Additionally, the levels of these miRNAs were significantly decreased after surgery, radiotherapy, and chemotherapy when compared to pre-treatment levels, indicating a role for these miRNAs as response biomarkers [59]. Heneghan et al. also demonstrated that circulating levels of miRNA-195 and let-7a were increased in breast cancer patients compared to controls [55]. miRNA-195 had an 85.5% sensitivity and 100% specificity for detecting patients with breast cancer, with let-7 having 77.6% and 100% sensitivity and specificity, respectively. Notably, tumour expression of miRNA-195 was significantly increased in stage IV cancers, compared to stages I and II disease, indicating a role of miRNAs in disease prognosis [55]. Furthermore, miRNAs have been shown to be valuable indicators of response to neoadjuvant chemotherapy, with Ohzawa et al. demonstrating differences in miRNA expression profiles between HER-2-positive patients achieving pathological complete response and non-pathological complete response to neoadjuvant chemotherapy and trastuzumab [60]. miRNA-210 predicted response to neoadjuvant chemotherapy including trastuzumab, with higher baseline levels prior to neoadjuvant chemotherapy associated with residual disease [61]. Davey et al. carried out a systematic review highlighting the ability of miRNAs to be used as biomarkers predictive of the response of early stage HER-2-overexpressing breast cancers to neoadjuvant chemotherapy and/or anti-HER-2 therapies, with 41 miRNAs showing increased expression in patients who responded to neoadjuvant therapy, and 29 miRNAs with reduced expression in patients who responded [62]. Bouz Mkabaah et al. performed a systematic review of the literature, assessing the role of miRNAs in predicting the recurrence of breast cancer, with a total of 44 miRNAs identified as being useful in this goal [25]. McGuire et al. showed that increased expression of miRNA-21 and miRNA-145 in whole blood samples was associated with non-response to neoadjuvant chemotherapy in luminal breast cancer. Additionally, reduced levels of miRNA-21 and miRNA-195 were found in patients who responded to neoadjuvant chemotherapy. These data highlight the exciting potential of miRNAs to help determine which patients will benefit from neoadjuvant chemotherapy and help avoid unnecessary side effects in patients unlikely to respond [63].

The role of miRNAs as biomarkers of cardiovascular disease is well studied, with numerous examples. miRNA-1 was shown to be decreased in serum samples of patients with symptomatic heart failure, depending on the New York Heart Association (NYHA) class. Furthermore, miRNA-21 (a traditional cancer-related biomarker) levels were found to be upregulated in patients with heart failure [64]. One study found that the ability of brain natriuretic peptide (BNP) in predicting heart failure improved when adding miRNA-221 or miRNA-328 to the analysis. They also found that adding a combination of miRNAs (miRNA-375, miRNA-328, miRNA-221, and miRNA-30c) to BNP testing aided in differentiating heart failure with reduced ejection fraction from heart failure with preserved ejection fraction, with an area under the curve of 0.854 [65]. The role of plasma miRNA-145 as a biomarker of the failing heart is postulated by its downregulation in patients with heart failure compared with controls. Its role in myocardial infarction is evident from the same study, in which lower levels of miRNA-145 were found in the plasma of patients who had suffered acute myocardial infarction [66]. Plasma miRNA-29b was found to be downregulated in patients with congestive heart failure or atrial fibrillation. Additionally, miRNA-150 was shown to be downregulated in atrial fibrillation patients [67]. This highlights the potential use of miRNAs as important biomarkers of arrhythmias [68]. 

## 4. Regulatory Functions of miRNAs in Breast Cancer

Cancer is an important cause of mortality, expected to rank as a leading cause of death worldwide this century [2]. This review will focus on the multiple roles of miRNAs in breast cancer specifically. Many functions of miRNAs in breast cancer have already been elucidated, such as their role in subtype classification [57]. The role of miRNAs in cancer development was strongly suspected when miRNAs were found to be located in close proximity to cancer-causing genes [69]. Additionally, miRNA-10b was found to be upregulated in metastatic breast tumours but not in primary tumours without metastasis, highlighting the ability of miRNAs to prognosticate disease [70]. miRNAs are involved in the regulation of gene expression at the post-transcriptional level [71], with complementarity affecting mRNA degradation versus gene silencing [72]. miRNAs in breast cancer development can be oncogenic or tumour-suppressive [73]. The upregulation of oncogenes diminishes the function of tumour-suppressive genes, while the downregulation of tumour-suppressive genes facilitates pathways allowing tumourigenesis [74]. These processes enhance the development of cancer hallmarks, including angiogenesis, growth and proliferation, metastasis, and evading cell death [75]. Mitchell et al. illustrated that tumour-derived miRNAs were present in the plasma of mice with prostate cancer, without an increase in overall miRNA quantities [53]. These small RNAs represent exciting tools for disease diagnosis and progression, given their presence and stability in body fluids [76].

miRNAs are believed to regulate up to one-third of all protein-coding genes [77]. As already mentioned, they regulate multiple processes such as cell proliferation and apoptosis [27]. The dysregulation of these processes can result in multiple disease states, ranging from cancers [78] to cardiovascular disease [79]. Breast cancer occurs due to abnormalities of proliferation in cells of the ducts and glands of the breast [73]. It can be subclassified based on gene expression profiles into Luminal A, Luminal B, Erb-B2, and basal-like [3]. A study by van Schooneveld et al. showed that miRNA expression in breast cancer patients differs from their healthy counterparts [80]. miRNA levels in breast tumours were overall decreased compared to the control group, with miRNA-299-5P and miRNA-411 being the most differentially expressed between cancer and control groups. miRNAs also play significant roles in regulating breast cancer progression, with Xiaojian et al. demonstrating that decreased concentrations of miRNA-106b are associated with increased levels of matrix metalloproteinase 2, a collagenase shown to be involved in bone metastasis in breast cancer [81]. Ohzawa et al. discovered the usefulness of miRNAs as adjuncts in determining clinical outcomes and responses to neoadjuvant chemotherapy in HER-2-positive breast cancer [60]. Table 1 below highlights a number of miRNAs with identified involvement in breast cancer.

Cellular proliferation is noted as an important feature of cancer development, including breast cancer [75]. A fine balance exists between cell proliferation and arrest. Normal cells generally stop proliferating and remain quiescent after a set number of cell divisions [82]. miRNAs are emerging as important regulators of cellular proliferation, acting on numerous mediators such as cyclins and protein kinases [73]. Cyclin E1 is an important gene involved in cell proliferation, namely, at the G1/S phase transition, and can be impacted upon by miRNAs [83]. Luo et al. demonstrated the role of miRNA-15a in breast cancer, where it is downregulated compared to normal tissue (tumour suppressor). Additionally, miRNA-15a suppresses proliferation via the regulation of Cyclin E1 and acts to inhibit cell migration [83]. Furthermore, Huang and Lyu showed the function of another miRNA in cell proliferation. miRNA-483-3p upregulation was effective in preventing breast cancer cells from entering the S phase of the cell cycle, with Cyclin E1 again being the target (tumour suppressor) [84]. Zhou et al. showed the tumour-suppressive qualities of miRNA-143 in breast cancer, where overexpression reduced cell viability, with *ERK5* and *MAP3K7* being likely target genes [85]. 

miRNAs with oncogene-type properties have been identified; for example, miRNA-1207-5p was shown to promote breast cancer cell growth in one study by Yan et al., where its overexpression resulted in the downregulation of cell cycle suppressors *CDKN1A* and *CDKN1B* via the *STAT6* gene [86]. Shen et al. displayed the oncogenic properties of miRNA-492, which is overexpressed in breast cancer cell lines, resulting in decreased levels of *SOX7*, a tumour suppressor transcription factor involved in the Wnt/*B*-catenin pathway [87,88], leading to increased levels of proliferation via the upregulation of Cyclin D1, Rb phosphorylation, and c-Myc [89]. 

**Table 1 ijms-24-12694-t001:** Example miRNAs in breast cancer, with molecular targets highlighted.

miRNA	Expression	Targets	Reference
miRNA-10b	Upregulated	HOXD10	[70]
miRNA-299-5p	Downregulated	STK39	[80,90]
miRNA-411	Downregulated	Specificity Protein 1 (SP1)	[80,91]
miRNA-106b	Downregulated	Matrix Metalloproteinase 2 (MMP2)	[81]
miRNA-15a	Downregulated	Cyclin E1	[83]
miRNA-492	Upregulation	SOX7	[89]
miRNA-1207-5p	Upregulated	STAT6, CDKN1A, CDKN1B	[86]
miRNA-21	Upregulated	LZTFL1	[92]

## 5. Implications of Breast Cancer Therapies on Cardiovascular Function (miRNAs as Biomarkers)

Breast cancer is treated using a multimodality approach. Patients are treated using a combination of surgery, chemotherapy (either in a neoadjuvant or adjuvant setting), endocrine therapy, immunotherapy, monoclonal antibodies, and radiotherapy. These therapies are often associated with adverse cardiovascular outcomes [12,93,94]. With the 15-year relative survival rates for breast cancer approaching 80%, it is vitally important to consider the burden these treatments have on quality of life and non-cancer-related mortality in breast cancer patients [13,95]. A possible link between genes involved in breast cancer and those in cardiovascular disease may exist [96,97]. Cardiotoxicity is a major concern when prescribing chemotherapy [98]. Chemotherapy for breast cancer treatment is well described to have adverse cardiotoxic side effects, such as doxorubicin-induced heart failure [93]. Chemotherapy-related cardiac dysfunction is classified into type 1 related to anthracyclines and type 2 related to trastuzumab [99]. Anthracyclines constitute a widely prescribed chemotherapeutic agent in breast cancer management. They have a myriad of cardiovascular side effects, including abnormal heart rhythms, pericarditis, and myocarditis [100]. One meta-analysis found a greater than fivefold increase in congestive heart failure rates in patients treated with anthracyclines compared with non-anthracycline-based chemotherapy [101]. Taxanes have also been shown to have considerable cardiac side effects. Martin et al. randomised patients with node-positive breast cancer to adjuvant treatment with docetaxel plus doxorubicin and cyclophosphamide (TAC) or fluorouracil plus doxorubicin and cyclophosphamide (FAC) and found that there was a 1.6% incidence of heart failure in the TAC group compared to 0.7% in the FAC group (*p*-value = 0.09), indicating a potential increase in heart failure with docetaxel treatment [102]. Cyclophosphamide, an alkylating agent, is widely used in breast cancer management [103]. One retrospective analysis of 84 transplant patients treated with cyclophosphamide found that 17% of these patients developed congestive heart failure within 10 days of treatment [104]. A case report by Birchall et al. showed the association of haemorrhagic myocarditis with cyclophosphamide therapy [105]. 

The HER-2/*neu* gene is overexpressed in approximately 16% of stages I, II, and IIIa breast malignancies [106]. Trastuzumab is an effective agent for HER-2/*neu*-positive patients, with a 50% reduction in the recurrence rate observed post-completion of standard therapy [5]. Another study indicated a 5-year overall survival of 90.4% in the control group, compared with 100% in the adjuvant trastuzumab group in patients with small, node-negative, HER-2/*neu*-positive disease [107]. It should be noted that only one patient in the control group received adjuvant chemotherapy compared to all women in the trastuzumab group. Its cardiotoxicity is well documented, with one large Italian study indicating an over fourfold increase in congestive heart failure and cardiomyopathy in women treated with adjuvant trastuzumab-based therapy compared to standard chemotherapies in the first year of treatment [13]. 

In addition to chemotherapies, radiotherapy plays a significant role in the cardiotoxicity of breast cancer treatments. The risk of ischaemic heart disease was shown to increase by 7.4% for each 1 Gray increase in radiation exposure to the heart following radiotherapy for breast cancer [12]. The prospective Pathways Heart Study examined 14,942 women with a breast cancer history compared to 74,702 women without breast cancer. Patients in this study who were exposed to left-sided radiation therapy or endocrine therapy had higher rates of incident hypertension compared to controls [108].

We have already mentioned the role of miRNAs in predicting response to chemotherapy [60], but can miRNAs be useful in predicting cardiotoxicity? One systematic review identified a range of miRNAs as markers of cardiac damage in breast cancer patients treated with chemotherapy, highlighting the important role of these molecules in side effect management [109]. Traditional biomarkers of cardiac function include natriuretic peptides [110] and troponin levels [111]. One meta-analysis found an increase in troponin levels in cancer patients after chemotherapy when compared to pre-chemotherapy levels, with troponin serving as an indicator of systolic function [112]. miRNAs have also been implicated in cardiovascular disease [113]. One meta-analysis by Cheng et al. showed the potential of miRNA-499 to be used as a biomarker of myocardial infarction [114]. Freres et al. performed a prospective study evaluating the levels of cardiac-related circulating miRNAs in patients after anthracycline neoadjuvant chemotherapy for breast cancer. They found an increase in congestive-heart-failure-associated miRNAs after chemotherapy, including elevated levels of miRNA-126-3p, miRNA-199a-3p, and miRNA-423-5p [115]. miRNA-423-5p was previously found to be increased in heart failure patients compared to controls [116]. Rigaud et al. demonstrated a significant association between circulating levels of miRNA-1 and a reduction in left ventricular ejection fraction in female breast cancer patients treated with doxorubicin [117]. Additionally, cardiac-related miRNA expression, such as miRNA-29, has been found to be altered by radiotherapy, indicating a role of miRNAs in predicting radiotherapy-induced cardiotoxicity [118,119]. miRNA-574-3p was found to be elevated in serum after thoracic radiotherapy, with levels one standard deviation above the mean being associated with a 1.85 hazard ratio for grade 3+ radiation-induced cardiotoxicity [120].

## 6. Implications of Breast Cancer Therapies on Diabetes Mellitus

Diabetes mellitus represents one of the most common diseases worldwide, with a growing burden and is predicted to affect up to 7.7% of the world’s population by the year 2030 [121]. The consequences of DM are many-fold, with long-term adverse outcomes split into microvascular and macrovascular complications. Microvascular complications include nephropathy, neuropathy, and retinopathy, while macrovascular complications include premature cardiovascular and cerebrovascular morbidity, such as myocardial infarction and stroke [122]. Studies have highlighted an up to 20% increase in breast cancer risk in women with type 2 diabetes mellitus [123]. Insulin levels are increased for a long period surrounding disease onset [123]. Chappell et al. demonstrated a mitogenic effect of insulin on MCF-7 breast cell lines, further supporting this concept [124]. Additionally, the Pathways Heart Study found an increased risk of diabetes in breast cancer patients compared with controls, with this association increasing in groups receiving chemotherapy, left-sided radiation therapy, or endocrine therapy [108]. Breast cancer also shares miRNA involvement with diabetes, with miRNA-375 impacting insulin secretion [125], while also being shown to have tumour-suppressing qualities, being downregulated in malignant cells [126].

The use of hormonal therapies has been shown to increase the risk of developing diabetes in numerous studies. In a post hoc analysis, tamoxifen was shown to reduce insulin sensitivity in overweight women at high risk or diagnosed with breast cancer by nearly sevenfold [127]. Another nested case-control study followed over 14,000 breast cancer survivors aged ≥65 years of age and found that tamoxifen use was associated with an increased risk of developing diabetes [128]. This association was found only for recent discontinuation or current tamoxifen prescription and not when previous tamoxifen users were compared to women who had not received tamoxifen.

The Pathways Heart Study recently demonstrated an increased risk of diabetes in breast cancer patients, with a subdistribution hazard ratio (sHR) of 1.16 (adjusted) for all breast cancer patients compared to controls. When looking only at those patients who received any chemotherapy compared to controls, the sHR was 1.23 (adjusted), an increase for those women who received chemotherapy [108]. Conversely, one observational study provided evidence that pre-existing diabetes increases the risk of various chemotherapy-related side effects, including increased risk of hospitalisation for anaemia, neutropenia, and infections [129]. These important associations can potentially be predicted with miRNAs; unfortunately, to the best of our knowledge, a paucity of evidence exists on this topic.

## 7. Conclusions

In conclusion, this review highlighted the importance of miRNAs in breast cancer regulation, their use as biomarkers of disease, and explored the role of miRNAs in predicting treatment-induced toxicities, in particular, adverse cardiovascular side effects. miRNAs were found to be dysregulated following breast-cancer-related treatment [62], including miRNAs involved in cardiotoxicity [115]. Future directions include examining the therapeutic use of miRNAs in breast cancer to reduce toxicity to the host while maximising toxicity to cancer. Other exciting aspects, such as miRNA panels predicting breast cancer occurrence in high-risk women, may lead to the routine use of miRNAs in the clinical management of breast cancer [130]. Additional translational research efforts may further decipher the clinical utility of miRNAs as biomarkers to aid in the detection of treatment-induced toxicities to current therapeutic strategies in breast cancer and elicit the role of these molecules in predicting host response.

## Data Availability

Not applicable.

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
