# Peer review of "Biological Implications of MicroRNAs as Regulators and Biomarkers of Therapeutic Toxicities in Breast Cancer"

_ijms, 2023, doi:10.3390/ijms241612694_

Round 1
Reviewer 1 Report
Original Submission
Comments to Author:
Ms. Ref. No.: ijms-2535564
Title: Biological implications of MicroRNAs as Regulators and Biomarkers in Breast Cancer.
Minor Revision
Overview and general recommendation:
In the review, the authors explored the role of miRNAs in the onset of breast cancer and its role as candidate biomarkers. In general, the review was well organized and although the topic would be interesting to the readers of IJMS, several concerns need further clarification to ensure the suggestion for publication. I explain my concerns in more detail below.
Comments:
1. Since the review has focused on the role of miRNAs as biomarkers for breast cancer therapies toxicities, the title should be revised.
2. In the abstract, the word contemporary is repeated twice. Kindly change one.
3. I would suggest to include brief information about the 4 subtypes of breast cancer in the introduction.
4. Line 77: miRNA have been… change to: “miRNAs have”
5. Line 100: tumor-derived miRNAs
6. May be section 3 can be renamed to ‘miRNA as Biomarkers of Breast Cancer.’
7. Title for Table 1 should be renamed.
8. A table should be included to summarize different miRNAs in breast cancer.
Author Response
Dear Reviewer,
Many thanks for your feedback on our manuscript. Please see attached a copy with comments on changes made.
Kind regards,
Dr Raza Abbas Syed

Reviewer 2 Report
The Authors wrote a review on miRNAs in breast cancer, particularly focussing on their potential use as biomarkers and their involvement in cardiovascular disease and diabetes.
I personally think that the Authors should intensively revise their manuscript by avoiding repetition of concepts, following a logical order and a better division of the paragraphs, adding more focussed and recent references of original articles, and making tables more informative.
Here below, my point-to-point suggestions (besides a general revision, which is required):
Line 55: change or modify the word ‘contemporary’
Line 74: change ‘quantified’ with ’described’
Lines 74-75: please explain that this miRNA was not a human miRNA and cite the first discovered human microRNA.
Lines 77-78: too reductive take as examples only cell proliferation and apoptosis. Please be more generic or add a few more examples/references.
Line 80: specify the date in which the authors have access to the database
Line 81: ‘miRNA’ needs to be uppercase and plural
GENERAL: ATTENTION TO PLURAL FORMS OF MIRNA – many mistakes present – the English needs to be revised
Line 91: mRNA needs to be uppercase
Line 91-92: this concept has not been expressed properly. Please carefully revise it.
Line 93: remove ‘heavily’
96: I have never heard about the acronym ‘tsmiRs’. Please keep conventional abbreviations
Lines 93-103: Is this part properly placed? I am not sure this part should be in the miRNA biogenesis section.
REF. 47 (as an example): more recent literature needs to be cited. Please check latest publication on miRNA and their use as biomarkers.
Line 143: double spacing
GE-NERAL: ‘et al.’ should be in italic (it’s Latin)
Lines 184-194: I do not understand why Authors talk about Diabetes Mellitus since the review is focussed on breast cancer/cancer. Please, make more relevant references and more focussed.
Line 193: Please check the sentence with a ‘but’. It has no sense to me.
Line 207-209: classification of BC seems incomplete. Please revise.
Table 1: these are too few examples. Please make a more complete table.
Table 2: same for table 1. Moreover, the sentence ‘sample mirnas’ does not make sense to me. Please revise the subject.
Attention to uppercase words, to words which need to be in italic, and to plurals
Author Response

(The authors gave the same response as above.)
